# Optimization of Temporal Coding of Tactile Information in Rat Thalamus by Locus Coeruleus Activation

**DOI:** 10.3390/biology13020079

**Published:** 2024-01-28

**Authors:** Charles Rodenkirch, Qi Wang

**Affiliations:** Department of Biomedical Engineering, Columbia University, ET 351, 500 W. 120th Street, New York, NY 10027, USA; cr2771@columbia.edu

**Keywords:** locus coeruleus, temporal coding, feature selectivity, the ventral posteromedial nucleus of the thalamus (VPm), the whisker system

## Abstract

**Simple Summary:**

The locus coeruleus–norepinephrine (LC-NE) system plays a critical role in regulating various brain functions through its diffuse projections in the brain. However, how the LC-NE system modulates temporal coding of sensory information along the sensory pathway remains poorly understood. To address this question, we recorded single-unit activity from a rat’s somatosensory thalamus in response to whisker stimulation while stimulating the LC. By comparing the temporal structure of the thalamic responses to the same tactile stimulus with and without LC activation, we found that LC activation optimized the temporal coding of tactile signals in the thalamus by moving spikes to time points where they more accurately represent the tactile stimulus. The results shed light on the future development of neuromodulation technologies to enhance and/or restore brain functions.

**Abstract:**

The brainstem noradrenergic nucleus, the locus coeruleus (LC), exerts heavy influences on sensory processing, perception, and cognition through its diffuse projections throughout the brain. Previous studies have demonstrated that LC activation modulates the response and feature selectivity of thalamic relay neurons. However, the extent to which LC modulates the temporal coding of sensory information in the thalamus remains mostly unknown. Here, we found that LC stimulation significantly altered the temporal structure of the responses of the thalamic relay neurons to repeated whisker stimulation. A substantial portion of events (i.e., time points where the stimulus reliably evoked spikes as evidenced by dramatic elevations in the firing rate of the spike density function) were removed during LC stimulation, but many new events emerged. Interestingly, spikes within the emerged events have a higher feature selectivity, and therefore transmit more information about a tactile stimulus, than spikes within the removed events. This suggests that LC stimulation optimized the temporal coding of tactile information to improve information transmission. We further reconstructed the original whisker stimulus from a population of thalamic relay neurons’ responses and corresponding feature selectivity. As expected, we found that reconstruction from thalamic responses was more accurate using spike trains of thalamic neurons recorded during LC stimulation than without LC stimulation, functionally confirming LC optimization of the thalamic temporal code. Together, our results demonstrated that activation of the LC-NE system optimizes temporal coding of sensory stimulus in the thalamus, presumably allowing for more accurate decoding of the stimulus in the downstream brain structures.

## 1. Introduction

Our perception and cognition are heavily influenced by behavioral state, including arousal and attention [1,2,3,4,5,6,7,8]. It has long been established that major neuromodulatory systems in the brain, including the locus coeruleus–norepinephrine (LC-NE) system, play a critical role in regulating behavioral state [9,10,11,12,13,14,15,16,17,18,19]. Recent studies have also suggested that norepinephrine produced by the noradrenergic neurons of the LC exerts a strong influence on sensory processing, and therefore perception, through distinct noradrenergic receptors [20,21,22,23,24].

Most sensory information is relayed to the cortex through the thalamus [25,26,27,28]. In the whisker system, the thalamus is strategically placed to gate information flow to the cortex to create tactile perception and internally processes stimuli through the interplay between the ventral posteromedial nucleus (VPm) of the thalamus and the thalamic reticular nucleus (TRN) [29,30,31,32,33,34,35,36,37,38,39]. In vitro application of NE on brain slices results in the suppression of a resting leak potassium current that induces hyperpolarization in addition to the enhancement of the hyperpolarization activated cation current [40]. This creates a slight depolarizing shift that reduces the bursting firing for the dorsal lateral and medial geniculate neurons in the thalamus. A bursting event is multiple spikes occurring within short succession (i.e., inter-spike interval of 4 ms or less) and requires a relatively longer preceding hyperpolarized period (i.e., 100ms or greater) to prime the T-type calcium channels responsible for bursts [40,41,42,43]. An in vivo recording from VPm neurons during LC stimulation revealed that LC stimulation altered the strength of VPm neuron’s responses to the same whisker pad stimulation [21,22]. Using reverse correlation analysis, we have previously reported that LC activation enhanced the selectivity of the response of VPm neurons to specific features in whisker stimuli [20], resulting in an increase in the efficiency and rate of information transmitted about tactile stimuli. The increase in information transmission mainly resulted from LC regulation of intra-thalamic dynamics, while trigeminal neurons and cortico–thalamic feedback were not involved. Further supporting the notion that LC activation increases thalamic information transmission, LC activation also improved rats’ perceptual performance in a behavioral task where the rats were required to discriminate between two tactile features [20]. When the effects of NE in the thalamus were pharmacologically blocked, the benefit of LC stimulation on perceptual performance was abrogated, suggesting that the improvement in performance was mainly due to NE enhancement of thalamic sensory processing [20]. However, how the LC-NE system modulates the neural coding of sensory information, in particular, the temporal code and population code, remains not fully understood.

To address this question, we recorded single-unit response in the VPm to the repeated presentation of frozen white Gaussian noise (WGN) whisker stimulus with and without LC stimulation. In response to multiple presentations of the same whisker stimulus, VPm neurons responded reliably at specific time points with a dramatic, transient increase in their firing rate. These reliable and relatively temporally precise spike-evoking time points are termed events [44]. By comparing the temporal structure of VPm responses to repeated tactile stimulus with and without LC stimulation, we found that LC stimulation significantly altered the temporal structure of events within VPm responses. Specifically, it was found a substantial portion of events were removed during LC stimulation. Moreover, many new events emerged during LC stimulation. Through comparing the feature selectivity of spikes falling within events that were removed by LC stimulation to that of spikes falling within events that were newly emerged during LC stimulation, we found that spikes within the emerged events had a higher feature selectivity. This improved the accuracy of encoding, and therefore allowed the neurons to transmit more information about the tactile stimulus than they did without LC stimulation. The difference in feature selectivity arose as spikes within removed events were preceded by stimulus features within the WGN that less-closely matched the features the neuron selectively encodes when compared to spikes within emerged events. This suggests that the event removal and emergence process optimized the VPm temporal code in a manner that favors enhanced feature selectivity and information transmission. To functionally confirm that the optimized temporal coding of tactile stimulus during LC stimulation could allow for enhanced acuity of perception, we examined the ability to decode the original stimulus from an ideal observer point of view (i.e., only knowing the feature selectivity and spike train responses of the recoded population of VPm neurons). As expected, we found that reconstruction of the original stimulus from a population of VPm neurons’ spike trains and corresponding feature selectivity was more accurate with LC stimulation than without LC stimulation. Taken together, our results provide new experimental evidence demonstrating that the activation of the LC-NE system optimizes the temporal coding of sensory stimulus in the thalamus, allowing for more accurate decoding of the stimulus in the downstream brain structures.

## 2. Methods

All experiments involving animals were approved by the Institutional Animal Care and Use Committee (IACUC) at Columbia University. Some experimental data were published in a previous report [20]. However, different analyses of these data were performed in this study and new results are reported here.

### 2.1. Surgery and Electrophysiology

Surgical preparation for electrophysiological recording was published previously in detail [20,45]. Briefly, rats were anesthetized with sodium pentobarbital (30 mg/kg, intravenously, initial dose) that was maintained via syringe pump. Body temperature was maintained at 37 °C by a servo-controlled heating blanket (FHC). Vital signs and reflexes to aversive stimuli (toe or tail pinch) were closely monitored to index the depth of anesthesia. Rats were mounted to a stereotaxic frame to allow for craniotomies to be performed which gave access to the LC (~3.6-mm caudal to the Lamda, ~1.3 mm lateral from midline, ~5.4–6.0 mm ventral from the dura surface) and VPm (~3.3 mm caudal to Bregma, ~3.2 mm lateral from midline, ~5–5.7 mm ventral from the dura surface). For rats which underwent electrical LC microstimulation, a recording electrode (~2–3 MΩ, FHC Inc., Bowdoin, ME, USA) was advanced into the LC, with LC location being confirmed by the characteristic response of LC neurons to paw pinch [46]. The recording system was then disconnected and the electrode was connected to an electrical microstimulator (S88, Grass Instrument, Warwick, RI, USA). For rats that underwent optogenetic LC stimulation, 4 weeks prior to the experiment, a lentivirus was injected directly into the rats’ LC which allowed for selective transfection of noradrenergic neurons to express Channelrhodopsin2 (pLenti-PRSx8-hChR2(H134R)-mCherry, the UNC vector core, ~7 × 10^9^ vp/mL). At the beginning of an optogenetic LC stimulation experiment, a fiber optic cannula was advanced so as to be abutted to the LC, which was then attached to an LED driver during experimental sessions (Plexon, 493 nm wavelength, Dallas, TX, USA). We did not find systematic differences between results from electrical and optogenetic LC stimulation; both induced similar decreases in VPm burst spiking and increases in information transmission (Appendix A). Given both methods of LC activation produced functionally identical results, we combined the two data sets together in our analyses here. For all experiments, a tungsten recording electrode (~3–5 MΩ, FHC Inc., Bowdoin, ME, USA) was then advanced into the VPm, with VPm neurons being identified by their stereotaxic coordinates and the amplitude and temporal structure of their response to punctate whisker deflection [47].

For each VPm neuron, one of two different frozen blocks of WGN whisker deflection was repeatedly delivered to the principal whisker via a custom modified galvomotor [48] (galvanometer optical scanner model 6210H, Cambridge Technologies, Tullamarine, VIC, Australia) controlled by a closed-loop system (micromax 67145 board, Cambridge Technologies). For each VPm neuron, multiple whiskers were stimulated and the whisker that evoked the strongest response was identified as the principal whisker of the neuron. Single-unit recordings of VPm neurons’ responses to multiple repetitions of the same WGN stimulus were then captured via a Plexon recording system (OmniPlex, Plexon Inc., Dallas, TX, USA). During each recording, responses of the same VPm neuron to the same frozen WGN stimulus were recorded under two LC stimulation conditions (i.e., 2 Hz and 5 Hz stimulation frequency; electrical stimulation: 60 µA biphasic pulses with 200 µs per phase; optogenetic stimulation: 493 nm pulses with 5 ms duration; 20 mW/mm^2^) and without LC stimulation. Since our previous results showed that 5 Hz LC stimulation had a stronger effect on thalamic feature selectivity than 2 Hz LC stimulation, in this study we only analyzed the response of VPm neurons without LC stimulation versus their response with 5 Hz LC stimulation [20].

### 2.2. Reverse Correlation Analysis

Here, we modeled the response of VPm neurons using the linear-nonlinear-Poisson cascade model (LNP) [36,49]. By analyzing multiple responses of a neuron to the same frozen WGN stimulus, we can identify the kinetic features in the stimulus to which the neuron selectively responds (i.e., the linear stage of the model) and how selectively the neuron responds to those features (i.e., the nonlinear stage of the model). Here, we recovered each neuron’s significant features by first calculating the spike-triggered average (STA) followed by calculating the spike-triggered covariance (STC) matrix to recover the remaining set of significant features for any neurons which selectively responded to more than one kinetic feature [36,49].
$$STA=\frac{1}{N}\sum_{n=1}^{N}\vec{S}\left(t_{n}\right)$$
$$STC=\frac{1}{N-1}\sum_{n=1}^{N}\left[\vec{S}\left(t_{n}\right)-STA\right]{\left[\vec{S}\left(t_{n}\right)-STA\right]}^{T}$$
where *t_n_* is the time of the *n^th^* spike, $\vec{S}\left(t_{n}\right)$ is a vector representing the stimulus during the temporal window preceding that spike, *N* is the total number of spikes, and *^T^* denotes transpose. Statistical significance of features was determined using a bootstrap procedure [49]. To quantify the change in amplitude of features recovered during LC activation, we used a feature modulation factor previously defined as [20]:$$feature\;modulation\;factor=\frac{control\;feature\cdot feature\;during\;LC\;activation}{control\;feature\cdot control\;feature}$$

Once the linear portion of the LNP model was recovered, i.e., the kinetic features the neuron selectively responded to, we calculated the feature coefficients as the dot product between the neuron’s kinetic filter with the stimulus features preceding each spike. The probability distribution of feature coefficient values *k* given a spike (i.e., Prob(*k*|*spike*)) was subsequently generated from the feature coefficients. A probability distribution of all possible feature coefficient values (i.e., Prob(*k*)) in the stimulus was created and then calculated by sliding a 20 ms window through the entire 20 s WGN stimulus. Then the corresponding nonlinear-tuning functions for each feature were calculated as:$$Nonlinear\;tuning\;function=\frac{Prob(k|spike)}{\;Prob(k)}.$$
.
where *k*, the feature coefficient values, are the dot product between the linear filter and the preceding stimulus.

VPm neurons were stimulated in two directions across a fixed plane (e.g., both caudal and rostral directions). Some VPm neurons were found to have a directional preference for their kinetic feature (e.g., rostral displacement) while others responded to their kinetic feature delivered in either direction (e.g., rostral and caudal displacement). The strength of this directionality preference of the selective response to a specific feature was quantified by analyzing the symmetry of the nonlinear tuning function as follows:$$directionality\;alpha\;value=\frac{G\left(B\right)-G(-B)}{G\left(B\right)}$$
where *G* is the nonlinear tuning function and *B* is equal to 2 standard deviations of feature coefficient value.

Information conveyed by VPm neurons about the features they selectively responded to was quantified as [36,50]:$$Info\left(k;spike\right)=\int dk*Prob\left(k|spike\right)*\log_{2}\left(\frac{Prob\left(k|spike\right)}{Prob(k)}\right)$$
where *k* is the feature coefficient and the resulting bits/spike value indicates the mutual information between the absence/presence of that kinetic feature in the stimulus and the occurrence of a spike by this neuron.

To allow for identification of reliable events in the responses of neurons to the same WGN whisker stimulus, the peristimulus time histogram (PSTH) of each neuron’s responses was created by binning spikes (2 ms bins). The PSTH was then convolved with an adaptive boxcar kernel [44], whose size was dynamically increased from 1 at each bin until the bins spanned by that kernel contained at least 10 spikes, to produce a spike density function (SDF). A threshold (3 times the mean firing rate; using a threshold of 2 or 4 times the mean firing rate yielded qualitatively similar results, Appendix A) was then used to identify peaks in the SDF which were then considered events [44].

### 2.3. Analyzing Optimality of Sensory Encoding Event Time Points

To analyze how optimally a neuron encoded for the presence of a specific feature in a stimulus, we calculated the optimal temporal structure given the fixed number of events observed in the experimental recording. Further investigation of how event rate plays a role in optimal decoding of sensory stimulus is warranted but is beyond the scope of this project. Ideal event time points are the time points for which the stimulus immediately preceding this time point most closely matches the feature selectivity of the neuron. How closely the stimulus matches the feature selectivity at each time point is quantified by the feature coefficient vector, which is a vector of dot products between the encoded feature and a window of equal length slid along the stimulus. For directional neurons, ideal event time points then correspond to the largest positive peaks in the feature coefficient vector. For non-directional neurons, ideal event time points correspond to the largest magnitude peaks in the feature coefficient vector. Directionality was determined using the directionality alpha factor defined above. A threshold of alpha = 0.3 (threshold of 0.25 or 0.35 yielded similar results) was defined and feature selectivity with an alpha value below this threshold were considered to be non-directionally selective while those that fell above were considered to be directionally selective. The ideal time points were calculated separately for control and LC stimulation conditions.

### 2.4. Decoding VPm Responses

To reconstruct an approximation of the original stimulus from an ideal observer viewpoint, we first calculated the average temporal response pattern of each neuron to the incoming stimulus (e.g., the PSTH) as well as the features for which that neuron encoded. For neurons that were selective for multiple features, each feature–PSTH pair was considered unique. We then selected only the directionally selective feature–PSTH pairs to use for the initial reconstruction. This was carried out because, from an ideal observer viewpoint, the non-directionally selective features are not informative until directionality of the stimulus can be predetermined.

For each directionally selective feature–PSTH pair at each time point, the preceding strength of the feature present in the stimulus was assumed to be relative to the PSTH value in that bin (i.e., average spike count/trial at that time point). The reconstructed vector at each point for a directionally selective feature–PSTH pair was therefore calculated as:$${reconstruction}_{directional\;feature}\left(t\right)=\sum_{i=1}^{T-1}feature\left(T-i\right)*PSTH(t+i)$$
where the bin size for both the PSTH and feature are equal to the sampling frequency of the original stimulus (i.e., 5000 Hz, 0.2 ms bins) and *T* is the length of the feature. We then summed all reconstruction vectors corresponding to each directional feature–PSTH pair and took the z-score of the resulting vector to generate a reconstruction of the original stimulus.
$$directional\;reconstruction=z\;score(\sum {reconstruction}_{directional\;feature})$$

Using the directional reconstruction to approximate the original stimulus direction at any time point, we were then able to improve the reconstruction further by adding information from the non-directionally selective feature–PSTH pairs. To this end, for each non-directionally selective feature–PSTH pair, we generated a reconstruction which was at each point equal to:$${reconstruction}_{non-directional\;feature}\left(t\right)=\sum_{i=1}^{T-1}A*feature\left(T-i\right)*PSTH\left(t+i\right)$$
$$A=1\;if\;dot\left(directional\;reconstruction\left(t-T:t\right),feature\right)\geq 0$$
$$A=-1\;if\;dot\left(directional\;reconstruction\left(t-T:t\right),feature\right)<0$$
where the value of *A* effectively selects the directionality of the nondirectionl feature selectivity at any time point to be the direction which best matches the reconstructed stimulus generated from directional features only. Once we had calculated a reconstructed stimulus vector for each non-directionally selective feature–PSTH pair, we were then able to generate a reconstruction of the stimulus using both directional and non-directional feature–PSTH pair reconstructions as
$$completere\;construction=z\;score(\sum {reconstructions}_{directional\;feature}+\sum {reconstructions}_{non-directional\;feature})$$

To visualize how adding additional features improves accuracy of the reconstruction, each reconstruction was then recreated multiple times with each recreation adding filters in a random order while accuracy of the recreated stimulus was compared to the original after addition of each new filter.

### 2.5. Statistics

All statistical tests were two-sided. A one-sample Kolmogorov–Smirnov test was used to assess the normality of data before performing statistical tests. If the samples were normally distributed, a paired or unpaired Student’s t-test was used. Otherwise, the two-sided Mann–Whitney U-test was used for unpaired samples or the two-sided Wilcoxon signed-rank test for paired samples. Bonferroni correction was used for multiple comparisons.

## 3. Results

### 3.1. Reliable Response of VPM Neurons to Tactile Stimulus

We recorded the single-unit activity of VPm neurons in response to whisker stimulus composed of repeated frozen white Gaussian noise (WGN) patterns while varying the activation condition of the LC-NE system in pentobarbital-anesthetized rats (Figure 1a). In response to multiple presentations of the same WGN whisker stimulation, VPm neurons responded reliably at specific time points, which presumably correspond to sections of the stimulus which closely match the kinetic features for which the neuron selectively encodes. Once multiple responses of a neuron to the same frozen WGN stimulus had been recorded, a spike density function (SDF) was generated by first collapsing the peri-event raster into a peri-stimulus time histogram (PSTH), then smoothing the PSTH by convolving it with an adaptive kernel (see Section 2. Methods) [44]. Elevated firing rates at certain time points in the SDF are called events. Events within the SDF were then identified through using a threshold of 3× mean firing rate [44] (Figure 1b). We found that a substantial portion (34.4 ± 2.6%, mean ± S.E.M unless otherwise indicated) of spikes were within events.

To compare the functional difference between the spikes within events and outside events, we employed the linear-nonlinear-Poisson cascade model to examine the encoding of the high dimensional spatiotemporal tactile signals into a spike train by these neurons [36,49] (Figure 1c). Through reverse correlation analysis, we were able to recover the kinetic feature(s) to which each VPm neuron selectively responded and then calculate the corresponding nonlinear tuning function(s), which illustrate the sensitivity of the neuron’s response to how closely the stimulus resembles that feature. We were then able to use an information theoretic approach (see Section 2. Methods) to quantify the mutual information between a neuron’s spike response and the absence/presence of the features in the stimulus, for which the neuron selectivity encodes [20]. As expected, the spikes within events were more efficient in transmitting information about the encoded kinetic features as information transmission per spike for spikes within event was approximately 2.65 times higher than those outside events (Figure 1d–f, 0.37 ± 0.05 vs. 0.14 ± 0.024 bits/spike, *p* < 3.6 × 10^−10^, Wilcoxon signed-rank test; 32 neurons from 19 animals), suggesting the existence of temporal code in the VPm.

### 3.2. LC Stimulation-Modulated Temporal Structure of VPm Response to the Same Stimulus

We next examined the extent to which LC activation modulates thalamic temporal coding by comparing the temporal structure of events each VPm neuron used to encode tactile stimulus with versus without LC stimulation. When we overlay the peri-event raster and SDF of the VPm response with LC stimulation over that of the VPm response without LC stimulation, we found a clear change in the temporal event structure (Figure 2a). Some events are conserved across both control and LC-activation conditions (Figure 2a, purple boxes). However, we also observed that LC activation resulted in the removal of some events that were present under control conditions (Figure 2a, pink boxes). Further, LC activation resulted in the addition of some new events that were not present under control conditions (i.e., emerged events) (Figure 2a, green boxes). Moreover, the number of emerged events was smaller than the number of removed events (Figure 2b, 2.62 ± 0.18 vs. 19.2 ± 0.23 Hz, *p* < 0.027, paired t-test, 32 neurons from 19 rats), resulting in fewer events in VPm responses during LC activation than during the control condition (Figure 2c, 5.71 ± 0.43 vs. 5.0 ± 0.54 Hz, *p* < 0.026, paired t-test, 32 neurons from 19 rats). Approximately half of the events found during control conditions were removed with LC stimulation, while approximately 40 percent of the events found during LC activation were newly emerged and not present during control conditions (Figure 2d; removed events: 50.0 ± 3.6%, emerged events: 40.0 ± 3.3%, *p* < 4.6 × 10^−3^, paired t-test, 32 neurons from 19 rats), suggesting a major change in temporal structure. Consistent with our previous finding that LC activation reduces firing rate, here, we found fewer spikes within each event per trial during LC activation than under control conditions (Figure 2e, 0.61 ± 0.041 vs. 0.55 ± 0.031 Hz, *p* < 4.5× 10^−5^, paired t-test, 32 neurons from 19 rats). Since we have shown that LC activation increased information transmission for VPm neurons, this led us to hypothesize that LC activation may have optimized the thalamic temporal coding by removing less-optimal events and adding more-optimal events within VPm responses.

### 3.3. Spikes within Emerged Events during LC Activation Transmitted More Information Than Spikes within Removed Events

To quantitatively test this hypothesis, we evaluated feature selectivity and information transmission for spikes within the four types of events: spikes without LC stimulation that occurred during removed events, spikes without LC stimulation that occurred during conserved events, spikes during LC activation that occurred during conserved events, and spikes during LC activation that occurred during emerged events. When recovering the feature selectivity for each group of spikes, we found that the feature selectivity, measured as the feature modulation factor (see Section 2. Methods), of spikes within emerged events was much higher as compared to spikes within removed events (Figure 3a–c, 1.0 ± 0.1 for spikes within removed events vs. 1.7 ± 0.1 for spikes within emerged events, *p* < 6.2 × 10^−5^, paired t-test, 32 neurons from 19 rats). Consequently, spikes within emerged events transmitted more information about the encoded features than spikes within removed events, confirming that LC activation optimized temporal code in the thalamus (Figure 3d, 0.20 ± 0.02 bits/spike within removed events vs. 0.67 ± 0.10 bits/spike within emerged events, *p* < 3.9 × 10^−8^, Wilcoxon signed-rank test, 32 neurons from 19 rats). Interestingly, when comparing the coding property of spikes within conserved events during LC activation to spikes within conserved events during control conditions, the spikes during LC activation exhibited a higher feature selectivity (Figure 3e, 1.8 ± 0.1 for spikes within conserved events without LC stimulation vs. 2.2 ± 0.1 for spikes within conserved events with LC stimulation, *p* < 1.3 × 10^−6^, paired t-test, 32 neurons from 19 rats), and therefore greater information transmission (Figure 3f, 0.37 ± 0.05 bits/spike within conserved events without LC stimulation vs. 0.93 ± 0.14 bits/spike within conserved events with LC stimulation, *p* < 9.3 × 10^−10^, paired t-test, 32 neurons from 19 rats). This suggests that, even within conserved events, spikes are occurring relatively more often at time points more optimal to transmit stimulus-related information during LC stimulation. Taken together, these results indicated that LC activation optimized thalamic temporal coding of tactile stimulus by replacing spiking events at time points with relatively low information encoding efficiency with spiking events at time points with higher information encoding efficiency. Analyses in our prior studies suggest this action is probably mediated by norepinephrine regulation of T-type calcium channels in thalamic neurons as we have found that LC stimulation suppressed burst firing (i.e., multiple spikes with an interspike interval of 4 ms or less following a period of quiescence of at least 100 ms) for VPm neurons [20].

To examine the extent to which the effect of LC activation on the thalamic temporal code resulted from the reduction in burst firing during LC stimulation, we first deleted all bursting spikes from spike trains and then assessed the information transmission for spikes within removed events and spikes within emerged events. Similar to results from spike trains containing bursting spikes, the spikes within removed events transmitted less information than spikes within emerged events (Figure 4a, 0.28 ± 0.04 vs. 0.80 ± 0.12, *p* < 8.7 × 10^−8^, Wilcoxon signed-rank test, 32 neurons from 19 rats). This further analyses adds more evidence that it was not simply the NE-mediated removal of these bursting spikes that caused the LC-activation-induced increase in information transmission. Indeed, bursting spikes are suboptimal for information transmission, as a burst consists of multiple spikes of which only one is likely to be optimally temporally aligned to the feature. We see this reflected in the data as spikes in bursts carried less information than the average spike with and without LC stimulation (Figure 4b,c, without LC stim: 0.14 ± 0.025 vs. 0.08 ± 0.01 bits/spike, *p* < 1.3 × 10^−6^, Wilcoxon signed-rank test; with LC stim: 0.57 ± 0.12 vs. 0.37 ± 0.09 bits/spike, *p* < 4.1 × 10^−5^, Wilcoxon signed-rank test; 32 neurons from 19 rats). However, when bursting occurrences were considered each as a single spike, they carry significantly more information than the average spike (Figure 4b,c, without LC stim: 0.14 ± 0.025 vs. 0.17 ± 0.027 bits/spike, *p* < 3.5 × 10^−4^, Wilcoxon signed-rank test; with LC stim: 0.57 ± 0.12 vs. 0.71 ± 0.17 bits/spike, *p* < 3.2 × 10^−3^, Wilcoxon signed-rank test; 32 neurons from 19 rats), suggesting bursts themselves are not inherently uninformative (see Section 4. Discussion).

### 3.4. The Reorganization of the Temporal Structure of VPm Events during LC Activation Favors Ideal Event Placement

Having found that LC activation resulted in a temporal rearrangement of the event time points in each VPm neuron’s spiking response to the same WGN tactile stimulus clip, we next investigated how ideal the temporal arraignment of the event structure each VPm neuron used to encode the WGN stimulus clip was with and without LC stimulation. To answer this question, we first needed to consider, for a neuron with a specific feature selectivity, what an ideal temporal structure of its response would look like. Here, we confined our search for the neuron’s ideal response by using the same exact number of events in our ideal response as were present in the neuron’s actual SDF. To find the ideal time points for these events to occur to encode the most information about the presence of a selected feature, we first calculated the feature coefficient vector (i.e., dot product between the encoded feature and a window the same length slid through the stimulus). A very informative neuron would only respond at the time points when the feature coefficient had a large magnitude, e.g., the peaks in the resulting feature coefficient vector. It is important to note that the ideal time points depend on whether a neuron’s response is directionally sensitive to the sign of the feature coefficient (i.e., sensitive to only large positive feature coefficient values vs. sensitive to both large negative and positive feature coefficient values) (Figure 5a), and that directional selectivity varies across neurons. A neuron selectively responding to a specific feature in a directional fashion would ideally fire at large magnitude feature coefficients only if they are positive values (Figure 5b). In contrast, a neuron selectively responding to a specific feature in a non-directional fashion would ideally fire at large magnitudes of feature coefficients regardless of whether they were negative (the inverse of the feature) or positive (Figure 5c).

To determine whether a neuron’s feature selectivity was directional or non-directional, for each feature we quantified the directionality of the corresponding nonlinear tuning index using a directionality alpha value as defined by Petersen et al. [36] (see Section 2. Methods). A feature selectivity which is directionally selective will exhibit an asymmetric nonlinear tuning function (Figure 5a, left panel), and will have an alpha value close to 1. A feature selectivity which is not directionally selective will have a corresponding nonlinear tuning function that appears symmetric across the y axis (Figure 5a, right panel), and an alpha value close to 0. We found that LC stimulation slightly increased the directionality of the VPm neurons’ feature selectivity as measured by alpha values (Figure 5d, 0.44 ± 0.04 without LC stimulation vs. 0.55 ± 0.04 with LC stimulation, *p* < 6.8 × 10^−4^, paired t-test, 32 neurons from 19 rats). Therefore, when deciding if a neuron’s feature selectivity is directional or not, we used the average directionality alpha value between the feature selectivity with and without LC activation. Any resulting directionality value which fell beneath a threshold of 0.3 (see Section 2. Methods) was considered to be non-directionally selective while any that fell above was considered to be directionally selective.

After the directionality of each feature selectivity was calculated, we then identified the peaks in the corresponding feature coefficient which would be most ideal to position our events. Here, we conserved the same number of events as observed in the original response but moved these event times to be located at the peaks in the feature coefficient vector with the largest positive values for directionally selective features (Figure 5b, red stars) or largest absolute values for non-directionally selective features (Figure 5c, red stars). We were then able to compare these ideal event time points with the actual event time points observed with and without LC stimulation (Figure 5a,b, blue stars). Here, we found that LC stimulation increased the fraction of events that occurred at an ideal event time point (Figure 5e, 0.20 ± 0.01 without LC stimulation vs. 0.23 ± 0.01 with stimulation, *p* < 6.0 × 10^−6^, paired t-test, 32 neurons from 19 rats). Taken together, consistent with the previous results, this showed that LC activation resulted in a changing of the temporal event structure in such a way that favors more informative, and therefore more optimal, event locations.

### 3.5. LC Activation Improved Accuracy in Reconstructing the Original Tactile Stimulus from VPm Population Responses

Next, we assessed the functional consequences of improved information transmission during LC activation from a decoding perspective. Specifically, we examined how LC activation impacted the ability to decode the original whisker stimulus from the population response of VPm neurons [51]. To this end, we selected a subset of VPm recordings for which all the responses were driven by the same frozen WGN stimulus. We then analyzed, from an ideal observer standpoint of view, how accurately we could reconstruct the original stimulus knowing only the VPm neurons’ responses and feature selectivity (see Section 2. Methods). To reconstruct the original stimulus, we assumed the preceding strength of a feature in the stimulus was relative to the average spiking response at that time point of the neuron encoding that feature. To begin reconstruction, only directionally selective features were used as non-directional feature selectivity needed to be orientated correctly to improve reconstruction accuracy. Once a reconstruction using only directionally selective features was completed, we were able to then improve this reconstruction by adding information from non-directionally selective features. To do this, we used the direction of the stimulus reconstructed from only the directional feature selectivity at each time point to determine the direction of the non-directional feature selectivity.

As expected, we found that the final reconstruction was significantly more accurate when using the spiking response and feature selectivity of the neurons during LC stimulation than without LC stimulation (Figure 6a). This indicated that LC stimulation optimized the encoding of sensory-related information in the thalamus in a manner which allows for a more accurate recovery of the original stimuli from the thalamocortical spike trains. This suggests the accuracy of the perception of stimuli could be enhanced as well. Indeed, in our previous work, we found that LC stimulation enhanced the perceptual sensitivity of rats discriminating between two different frequencies of whisker stimulation [20]. We then wanted to quantify how LC stimulation affects how closely the reconstruction matches the original stimulus. Further, we were interested in how the number of VPm neurons used for decoding affected the accuracy of reconstruction of the stimulus. To this end, we performed the above method of decoding of the stimulus from directional PSTH–feature pairs multiple times for each possible number of features used. For each directional reconstruction, the order of features added was randomized and the correlation coefficient between that reconstruction and the original stimulus was saved. When looking at a plot of average correlation coefficient versus number of features used for directional reconstruction, we found, as expected, that the accuracy of the reconstruction increased with the increasing number of features used (Figure 6b). We also found that the more features used, the less additional accuracy adding another feature provides, indicating that there was some redundancy in the information carried by each of the encoded features. Importantly, we found that no matter how many features were used for reconstruction, LC stimulation resulted in a more accurate reconstruction as measured by either correlation coefficient (Figure 6b) or root-mean-square deviation (RMSE) between the reconstruction and original stimulus (Figure 6c). Interestingly, we observed that the difference between the accuracy of reconstruction with and without LC stimulation increased as the number of features decoded increased, suggesting that LC activation decreased the redundancy of information carried by VPm neurons. We further performed a similar analysis investigating how adding in different amounts of non-directional features improved the reconstruction accuracy (Figure 6b). The results of this analysis also showed that LC stimulation resulted in a more accurate reconstruction when decoded from both directional and non-directional features (Figure 6b,c).

## 4. Discussion

Previous work has suggested that sensory neurons could use temporal patterns of changes in firing rate to carry information about sensory signals [52,53,54,55]. Consistent with this, in the present study, our results showed that LC stimulation improved neural coding of sensory information by optimizing the temporal pattern of VPm responses to repeated presentations of frozen WGN whisker stimulation. It has been long recognized that the amount of information a sensory neuron transmits about a stimulus is mainly determined by the reliability and precision of the neuron’s response. However, we have previously shown that changes in the reliability and precision of the neuron’s response resulting from LC activation could not account for the improvement in information transmission that we observed experimentally [20]. Here, we have demonstrated, for the first time, that LC activation optimizes the temporal pattern of VPm responses by replacing suboptimal events containing less informative spikes with new events at different time points containing more informative spikes (Figure 3).

How sensory information is encoded by neural activity has long been a hotly debated issue. Initial experimental evidence showed neuronal firing rate was tightly correlated with stimulus strength (and later, other parameters), leading to the notion of rate code [56,57,58]. More recently, a growing body of evidence suggests a temporal code, which asserts that the timing of spikes also carries stimulus information [53,54,55,59,60,61,62,63,64,65]. This is particularly true when the synchrony between activities across a population of neurons is essential to achieve certain computations [45,66,67,68,69,70,71,72,73,74]. Although rate code and temporal code are not necessarily mutually exclusive, our results further support the existence of temporal code in the whisker thalamus by showing how the LC-NE system can optimize the temporal pattern of VPm responses to transmit more sensory information.

We have previously reported that LC stimulation in general improved the efficiency and rate of stimulus-related information transmitted by VPm neurons [20]. Here, we further showed the functional consequence of enhanced information transmission by demonstrating that LC stimulation allowed for a more accurate recovery of the original stimulus when decoding it from the response of a population of VPm neurons from an ideal observer viewpoint (Figure 6). This provides a mechanistic understanding about how LC stimulation may enhance the accuracy of the perception of sensory stimuli and suggests a pathway to employing neural stimulation technologies engaging the LC-NE system, including vagus nerve stimulation, to enhance sensory processing [75,76].

When investigating whether event time points occur at ideal locations, it must be considered that a VPm neuron may selectively encode for multiple features. Therefore, event time points which may be non-ideal for one of the neuron’s encoded features may be ideal for another. Interestingly, here, we found an increase in the fraction of events occurring at ideal times for neurons encoding a single feature as well as neurons encoding multiple features, when each of those features was analyzed individually. If the change in the temporal structure of events used to encode a whisker stimulus resulted in an improved feature selectivity for one feature at the cost of a degraded feature selectivity for another feature, we would expect to see a mixed effect of LC activation on the fraction of events at ideal times. Instead, we observed an improvement across the vast majority of individual features, suggesting that removed events were not ideal events for any of the features for which that the neuron encoded. In this way, LC activation-mediated changes in the temporal structure of the response does not shift the feature selectivity towards one feature at the expense of another (i.e., tune the neuron’s selectivity to a specific feature in the stimulus), but rather improves the feature selectivity for all features to which the neuron selectively responds.

The numerous physiological properties of the neuron, including absolute/relative refractory periods and the activation level of T-type calcium channels, impose influences on the temporal patterns of its response to sensory signals because these properties affect the probability of the neuron firing its next spike at successive time points. Previous work has shown that T-type calcium channels in the thalamus play an important role in LC modulation of thalamic sensory processing [20]. Although burst firing is an indicator of T-type calcium channel activation, our results here present the nuanced finding that the removal of bursting spikes contributed to the NE-mediated enhancement of information transmission but cannot account for the majority of the improvement in temporal coding. Yet our earlier studies strongly suggested that the effect of LC-NE activation on thalamic sensory processing resulted from NE regulation of T-type calcium channel activation. Together, these findings suggests NE suppression of T-type calcium channels not only reduces burst firing, but also removes the continuous fluctuating influence of T-type calcium channels on thalamic relay neurons’ membrane potential. Supporting this hypothesis, our previous modeling work demonstrated that LC activation reduces metrics that correlate with cell membrane potential fluctuations. The fluctuating influence of T-type calcium channels does not specifically encode information related to incoming sensory stimuli so when removed should potentially allow the response of VPm neurons’ to be more correlated with sensory-related inputs from the brainstem principal trigeminal nucleus and TRN. Increasing the correlation of VPm neurons’ responses to relevant sensory information input could then allow these neurons to respond with spikes more optimally at time points where the stimulus features most closely match the feature it encodes. Because strong inhibition from TRN neurons is necessary for T-type calcium channels in VPm neurons to be activated [77,78,79,80,81], future work to investigate LC modulation of TRN responses to whisker stimuli is warranted to shed light on the effect of complex VPm-TRN interplay on thalamic temporal coding of tactile signals.

## 5. Conclusions

Our data provided new experimental evidence demonstrating that LC activation induces the rearrangement of the temporal structure of thalamic responses to a repeatedly delivered WGN tactile stimulus. The rearrangement replaced the spikes transmitting less information about the kinetic features present in the stimulus with spikes transmitting more information, indicating that LC activation optimizes the temporal coding of sensory information in the thalamus.

## Figures and Tables

**Figure 1 biology-13-00079-f001:**
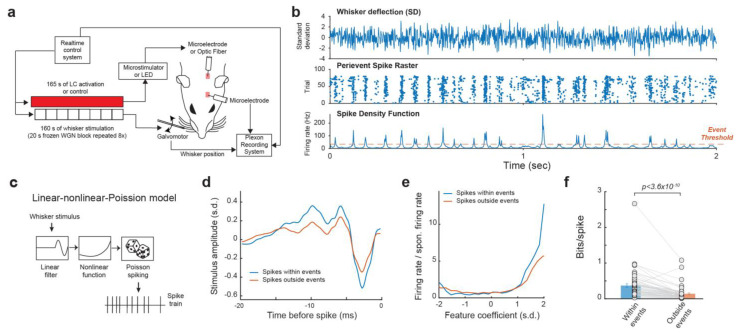
Reliable response of VPM neurons to tactile stimulus. (**a**) Experimental set up. (**b**) Example raster plot and spike density function of VPm response to repeated presentation of a tactile stimulus. (**c**) Linear-nonlinear-Poisson cascade model. (**d**,**e**) Feature selectivity of spikes within and outside of events. (**f**) spikes within events transmit more information about tactile stimulus than spikes outside of events.

**Figure 2 biology-13-00079-f002:**
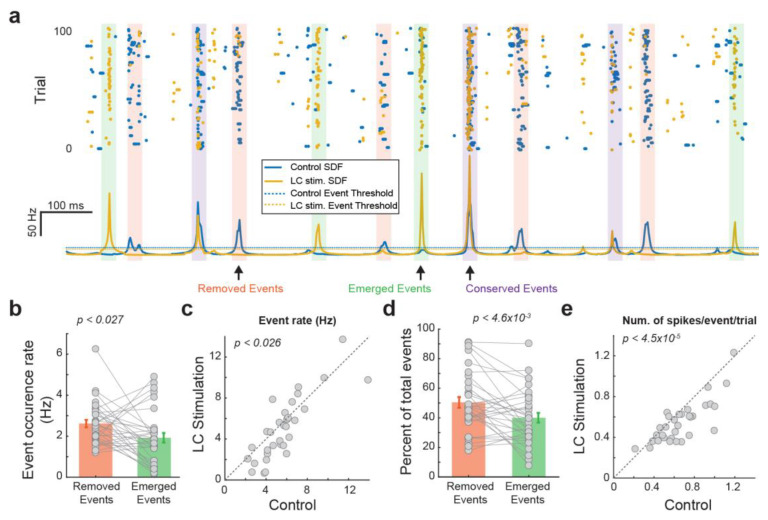
LC stimulation altered the temporal structure of VPm responses. (**a**) Characterization of different types of events of an example VPm neuron. (**b**) Rate of the removed events and emerged events. (**c**) LC activation decreased the number of events. (**d**) Percentage of the removed events and emerged events. (**e**) Number of spikes per event in a trial with and without LC stimulation.

**Figure 3 biology-13-00079-f003:**
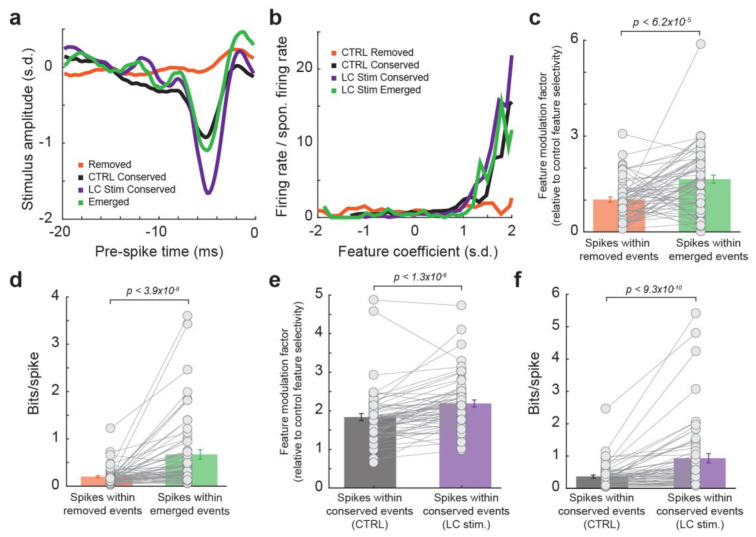
Emerged events during LC activation transmitted more information than removed events. (**a**) Example encoded feature by spikes within removed events (orange), conserved events without LC stimulation (black), conserved events with LC stimulation (purple), and emerged events (green). (**b**) Nonlinear tuning function for the spikes within the four types of events. (**c**) Feature modulation factor for spikes within removed events vs. spikes within emerged events. (**d**) Information transmission for spikes within removed events vs. spikes within emerged events. (**e**) Feature modulation factor for spikes within conserved events without LC stimulation vs. spikes within conserved events during LC stimulation. (**f**) Information transmission for spikes within conserved events without LC stimulation vs. spikes within conserved events during LC stimulation.

**Figure 4 biology-13-00079-f004:**
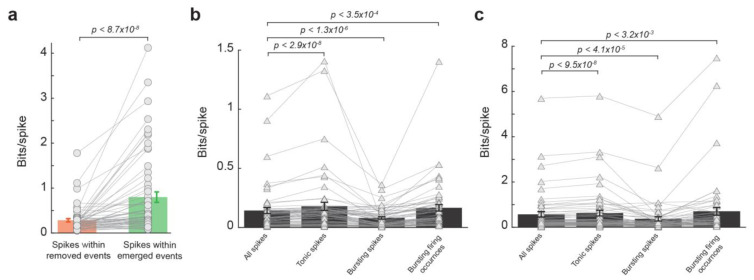
LC modulation of temporal code was not due to changes in burst firing. (**a**) Spikes within emerged events transmitted more information than spikes within removed events after bursting spikes were removed from the spike trains. (**b**) lnformation transmission per spike for all spikes, tonic spikes, bursting spikes, and burst firing occurrences without LC stimulation. (**c**) Information transmission per spike for all spikes, tonic spikes, bursting spikes, and burst firing occurrences with LC stimulation.

**Figure 5 biology-13-00079-f005:**
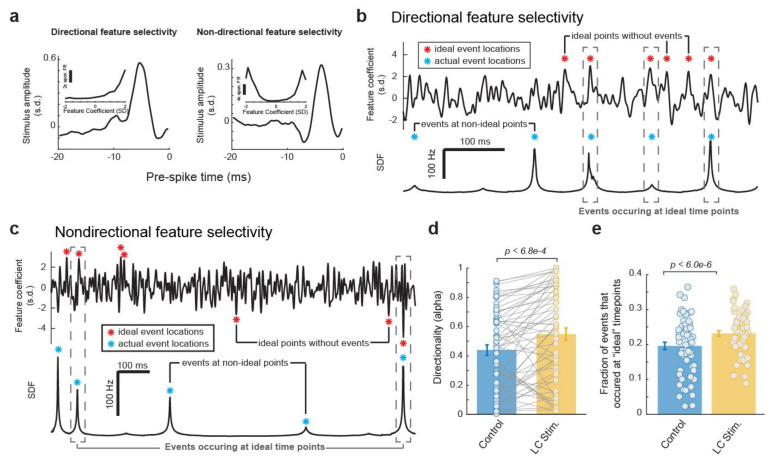
The reorganization of the temporal structure of VPm events during LC activation favors ideal event placement for optimal coding of stimulus. (**a**) Example directional and non-directional feature selectivity. (**b**) Plots of feature coefficient (top) and SDF (bottom) from an example VPm neuron with directional feature selectivity illustrating ideal time points for events to occur. (**c**) Plots of feature coefficient (top) and SDF (bottom) from an example VPm neuron with non-directional feature selectivity illustrating ideal time points for events to occur. (**d**) LC activation increased the directionality of VPm feature selectivity. (**e**) LC activation increased the fraction of events that occurred at an ideal event time points.

**Figure 6 biology-13-00079-f006:**
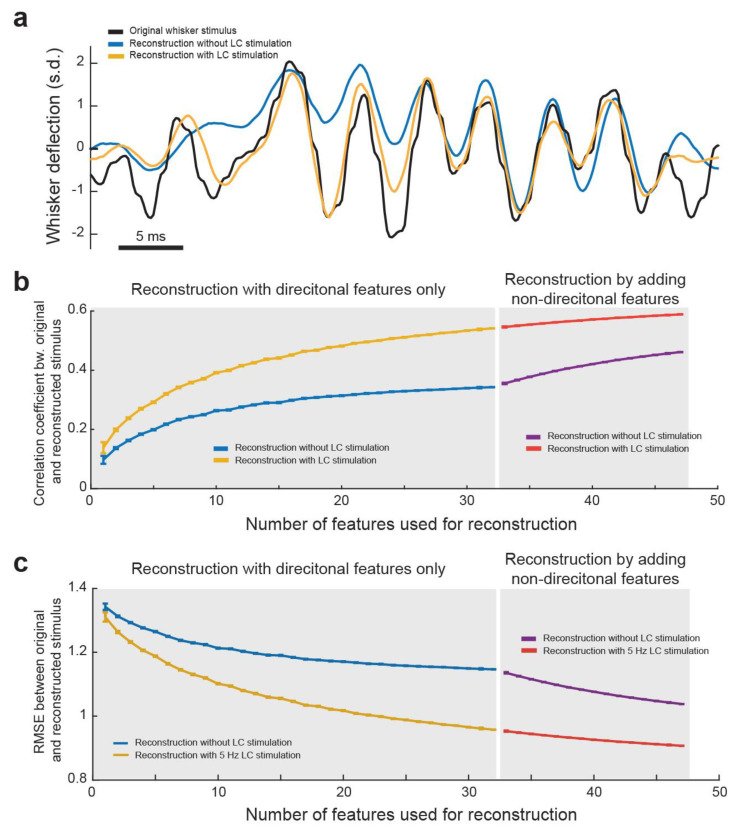
LC activation improved accuracy in reconstructing the original tactile stimulus from VPm population responses. (**a**) Example plot showing the original whisker stimulus, reconstructed stimulus without LC stimulation. and reconstructed stimulus with LC stimulation. (**b**) The correlation coefficient between the original and reconstructed stimulus with and without LC stimulation. (**c**) Root-mean-square deviation between the original and reconstructed stimulus.

## Data Availability

The data that support the findings of this study are available from the corresponding author upon request.

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
