# Peer review of "Optimization of Temporal Coding of Tactile Information in Rat Thalamus by Locus Coeruleus Activation"

_biology, 2024, doi:10.3390/biology13020079_

Round 1

Reviewer 1 Report

Comments and Suggestions for Authors

In this study, the authors extended their previous work by examining the effect of LC activation on the temporal code of the VPm thalamus in response to repeated white Gaussian noise stimuli. The experimental design is sound, the analyses are properly performed, and the results are very interesting. I believe it will be a valuable and timely addition to the literature.

I have only a few comments:

1. It looks like LC activation significantly reduces temporal jitter across trials in Figure 2a (narrower yellow SDFs within events). Maybe I missed something, but did the information metric in Figure 3d capture this? If so, how? If not, it would be great to add this analysis as it is an even more direct measure of temporal coding.

2. Can you explain how you chose the amplitude of the WGN stimuli? It seems that the stimuli were near threshold. I wonder if the effect of LC stimulation is only apparent in this regime, i.e., a region with a lot of non-linearity in the cell's input-output curve. For example, will the conclusions still hold if stronger or more natural stimuli are used?

3. Line 279: The orange boxes look like pink boxes to me.

Author Response

Reviewer 1

In this study, the authors extended their previous work by examining the effect of LC activation on the temporal code of the VPm thalamus in response to repeated white Gaussian noise stimuli. The experimental design is sound, the analyses are properly performed, and the results are very interesting. I believe it will be a valuable and timely addition to the literature.

Thank you for your support.

I have only a few comments:

  1. It looks like LC activation significantly reduces temporal jitter across trials in Figure 2a (narrower yellow SDFs within events). Maybe I missed something, but did the information metric in Figure 3d capture this? If so, how? If not, it would be great to add this analysis as it is an even more direct measure of temporal coding.

We apologize for the confusion. We have actually calculated precision of events (i.e. temporal jitter) with and without LC activation in our previous work and found that LC activation did not significantly increase the precision of responses across neurons (Supplementary Fig. 8 in Rodenkirch et al., Nat. Neurosci. 2019). Figure 2a is an example neuron that happened to have improved precision during LC activation.

  1. Can you explain how you chose the amplitude of the WGN stimuli? It seems that the stimuli were near threshold. I wonder if the effect of LC stimulation is only apparent in this regime, i.e., a region with a lot of non-linearity in the cell's input-output curve. For example, will the conclusions still hold if stronger or more natural stimuli are used?

We chose the amplitude of the WGN stimuli based on methods previously shown to work well for reverse correlation analysis of the vibrissa pathway. Specifically, Petersen et al Neuron 2008 – “Diverse and temporally precise kinetic feature selectivity in the VPm thalamic nucleus”. In that paper, the WGN noise stimulation employed has a standard deviation of 70 um occurring to the whisker 3mm from the skin, resulting in roughly 1.3 degrees of deflection standard deviation for the stimulus. In our study we used two different WGN clips, computationally generated by MATLAB the same way, one with an SD of 1.2 degrees and the other with a SD of 1.4 degrees, and both produced similar results. We agree though that our investigation was within a limited range of stimuli and further studies investigating effects of LC stimulation on other types of stimulus are warranted.

  1. Line 279: The orange boxes look like pink boxes to me.

Thank you for catching this. The typo was corrected.

Reviewer 2 Report

Comments and Suggestions for Authors

In their manuscript entiteled „Optimization of Temporal Coding of Tactile Information in the Thalamus by Locus Coeruleus Activation“ Charles Rodenkirch and Qi Wang investigated in detail how a stimulation of the Locus Coeruleus influences the thalamic processing of tactile information. Their main observation is that the representation of a stimulation vector representing Gaussian noise by the spike patterns of VPm neurons was significantly changes. By a quantification of stimulus features leading to spikes, the authors revealed that this change in the response pattern was caused by the disappearance of spike responses carrying less information, while new spike responses with a high information content emerged upon LC stimulation, resulting in a higher information content in the spiking activity. Accordingly, the reconstruction of the whisker stimulus from the spiking activity was more accurate using the spikes recorded under LC stimulation. Finally, the authors also found evidences that this increase in information transfer was not merely due to the reduction of bursting activity upon LC stimulation.

The present manuscript was based on a previous publication by the authors (Charles Rodenkirch, Yang Liu, Brian J Schriver, Qi Wang (2019) Locus coeruleus activation enhances thalamic feature selectivity via norepinephrine regulation of intrathalamic circuit dynamics, Nat Neurosci. 2019 22:120-133) using part of the already published rexperiments. However, the result in the present manuscript did not overlap with this publication and the analysis included in the present manuscript nicely augments the conclusions about the role of LC activity for sensory information processing.

The manuscript is clearly written with a clear introduction, a good description of the experimental findings and a fair discussion. However, in my opinion the rather complicated analysis should be described in more detail in the materials part (see some of my minor comments).

I recommend to accept the present manuscript after minor revisions.

Point 1 (Line 50): Please consider to describe in the introduction your previous finding, that the effect of LC stimulation was mediated mainly by alterations on the level of VPn neurons, while trigeminal neurons and corticothalamic feedback loops where not involved.

Point 2: Please consider to readdress the functional implication of the more reliable representation of tactile features in VPn activity upon LC stimulation. In the present manuscript this effect was considered as “beneficial effect” (Line 56), “enhance the accuracy of the perception of sensory stimuli“ (line 483)  or “[improvement] of the feature selectivity for all features the neuron selectively responds to” (Line 500). However, this view raises the question why VPn neurons not always uses this “high-reliability mode”. Can you speculate about the advantages of the non-adrenergic mode in the non-arroused VPn? Can the lower selectivity decrease the theshold of tactile detection on the drawback of reduced spatiotemporal tactile resolution? Is it possible that both “modes” of VPn transfer functions represent distinct, but behavioral relevant functional states.

Point 3: Please consider whether you can provide a more specific aim then “how the LC-NE system modulates neural 61 coding of sensory information remains not fully understood”.

Point 4: “Once the linear portion of the LNP model was recovered, i.e. the kinetic features the neuron selectively responded to,…” (Line 152). Here and in several other instances (eg. Line 259) you explicitly conclude that the features represent the kinetic properties of the WGN stimulus. Please consider to discuss why you can exclude that the mere amplitude of the stimulus dominated the information content of the prestimulus interval.

Point 5: “Having found that LC activation resulted in a reconstruction of the temporal structure of VPm responses encoding the same tactile stimulus, ….” (Line 354). To me this statement is maybe an overinterpretation of your observations or, at least, slightly misleading. I'm aware that you consider the complete WGN-sequence as “the tactile stimulus”, but I still found this statement misleading. Probably most researcher would identify a “tactile stimulus” as a single, maybe complex, whisker deflection. In principle, you mainly demonstrate a shift in the feature vectors related to a neuronal spiking event between control and LC- stimulated trials. Please consider to rephrase this statements. In addition, can you reveal from your datasets whether the temporal code of a single “remaining event” representing an identical stimulus feature is altered upon LC-stimulation?

Minor point 1 (Line 45): Please add a reference to this statement.

Minor Point 2 (Line 47): Please be more specific here, as there are distinct ion mechanisms leading to bust spiking in other neuron types.

Minor Point 3 (Line 55-60): Please consider to provide here clearer references to the relevant publication.

Minor point 4 (Line 99-101): Please be more specific here. Range of PB concentration, how do you monitor the anesthesia depth, approx. duration of the operation? Also add a statement about the means to minimize animal suffering.

Minor Point 5 (Line 107): Providing stereotactic parameters may be helpful here. Do you perform histology to identify the correct location and the ration of LC neurons transfected?

Minor Point 6 (Line 121): Please describe in more detail how you identify that the principal whisker was stimulated.

Minor point 7 (Line 131): Please add a reference here informing the reader that this information is a published result.

Minor point 8 (Line 145): I miss a description of the exponent “T”.

Minor Point 9 (Line 151): Please provide a definition for the tern “conditional feature”.

Minor Point 10 ( Line 160): I was misled by the term “directionality” at this point, assuming here that you also investigated the tuning map of the VPn neuron. Consider stating here that directionality just refers to forward-backward.

Minor Point 11 (Line 173): From Fig. 1 it seems that this deconvultion is symmetrical in time and may thus interfere with the precise detection of “event” onsets (as the deconvoluted SDF will depend on the maximal frequency). Please estimate whether this will substantially influence the feature detection.

Minor point 12 (Line 290): Sentence incomplete??

Minor Point 13 (Line 307): “we found that the feature selectivity [….] of spikes within emerged events was much improved as compared to spikes within removed events”. Please consider to use a comparative term like “higher”, as there can be no functional progression in these mutually exclusive events.

Minor Point 14 (Line 310): Shouldn't this be “about encoded features”, as the comparison between these events is not restricted to a predefined feature?

Author Response

Reviewer 2

In their manuscript entiteled „Optimization of Temporal Coding of Tactile Information in the Thalamus by Locus Coeruleus Activation“ Charles Rodenkirch and Qi Wang investigated in detail how a stimulation of the Locus Coeruleus influences the thalamic processing of tactile information. Their main observation is that the representation of a stimulation vector representing Gaussian noise by the spike patterns of VPm neurons was significantly changes. By a quantification of stimulus features leading to spikes, the authors revealed that this change in the response pattern was caused by the disappearance of spike responses carrying less information, while new spike responses with a high information content emerged upon LC stimulation, resulting in a higher information content in the spiking activity. Accordingly, the reconstruction of the whisker stimulus from the spiking activity was more accurate using the spikes recorded under LC stimulation. Finally, the authors also found evidences that this increase in information transfer was not merely due to the reduction of bursting activity upon LC stimulation.

The present manuscript was based on a previous publication by the authors (Charles Rodenkirch, Yang Liu, Brian J Schriver, Qi Wang (2019) Locus coeruleus activation enhances thalamic feature selectivity via norepinephrine regulation of intrathalamic circuit dynamics, Nat Neurosci. 2019 22:120-133) using part of the already published rexperiments. However, the result in the present manuscript did not overlap with this publication and the analysis included in the present manuscript nicely augments the conclusions about the role of LC activity for sensory information processing.

The manuscript is clearly written with a clear introduction, a good description of the experimental findings and a fair discussion. However, in my opinion the rather complicated analysis should be described in more detail in the materials part (see some of my minor comments).

I recommend to accept the present manuscript after minor revisions.

Thank you for your support.

Point 1 (Line 50): Please consider to describe in the introduction your previous finding, that the effect of LC stimulation was mediated mainly by alterations on the level of VPn neurons, while trigeminal neurons and corticothalamic feedback loops where not involved.

Thank you for your suggestions. We have included our previous findings in the introduction.

Point 2: Please consider to readdress the functional implication of the more reliable representation of tactile features in VPn activity upon LC stimulation. In the present manuscript this effect was considered as “beneficial effect” (Line 56), “enhance the accuracy of the perception of sensory stimuli“ (line 483)  or “[improvement] of the feature selectivity for all features the neuron selectively responds to” (Line 500). However, this view raises the question why VPn neurons not always uses this “high-reliability mode”. Can you speculate about the advantages of the non-adrenergic mode in the non-arroused VPn? Can the lower selectivity decrease the theshold of tactile detection on the drawback of reduced spatiotemporal tactile resolution? Is it possible that both “modes” of VPn transfer functions represent distinct, but behavioral relevant functional states.

Thanks for pointing this out. We agree with you that it is possible that LC stimulation enhances the discrimination capacity of the neurons at the cost of their detection performance. Indeed, previous studies by Sherman et al suggest bursts may be beneficial as a wake-up call in response to salient stimuli during inattentive states; see “A wake-up call from the thalamus, Nat. Neurosci., 2001”. From our findings, the influence of the calcium t-channels that allows for this bursting is unideal for accurate encoding and thus likely why during attentive states increased levels of NE suppress the calcium t-channel influence. Although “beneficial” effects of LC activation is subjective, the statement of “enhance the accuracy of the perception of sensory stimuli“ (line 483)  or “[improvement] of the feature selectivity for all features the neuron selectively responds to” (Line 500) were based on experimental evidence related to rate and efficiency of information encoded by spikes. We have clarified the beneficial effect in the revised manuscript.

Point 3: Please consider whether you can provide a more specific aim then “how the LC-NE system modulates neural coding of sensory information remains not fully understood”.

 Thank you. We have added the temporal code and population code to the sentence.

Point 4: “Once the linear portion of the LNP model was recovered, i.e. the kinetic features the neuron selectively responded to,…” (Line 152). Here and in several other instances (eg. Line 259) you explicitly conclude that the features represent the kinetic properties of the WGN stimulus. Please consider to discuss why you can exclude that the mere amplitude of the stimulus dominated the information content of the prestimulus interval.

Thank you for your suggestion. To clarify, the feature selectivity calculated here from experimental data is done so via a reverse correlation analysis that allows for recovery of any changes in displacement, velocity, acceleration, and further derivative of kinetic movement. Analyzing experimental data we found similar results as Petersen et al.; specifically, that neurons only responded to diverse and temporally precise kinetic features that were less than 20 ms in length. This suggests that when being stimulated with continuous WGN the response of these neurons is not largely influenced by any displacement prior than 20 ms prior to the spike.  We agree with you that future work is warranted to conclusively test the effect of stimulus amplitude prior to the encoded feature on how LC activation modulated temporal code which may be subtle and need a large number of samples to distinguish.

Point 5: “Having found that LC activation resulted in a reconstruction of the temporal structure of VPm responses encoding the same tactile stimulus, ….” (Line 354). To me this statement is maybe an overinterpretation of your observations or, at least, slightly misleading. I'm aware that you consider the complete WGN-sequence as “the tactile stimulus”, but I still found this statement misleading. Probably most researcher would identify a “tactile stimulus” as a single, maybe complex, whisker deflection. In principle, you mainly demonstrate a shift in the feature vectors related to a neuronal spiking event between control and LC- stimulated trials. Please consider to rephrase this statements. In addition, can you reveal from your datasets whether the temporal code of a single “remaining event” representing an identical stimulus feature is altered upon LC-stimulation?

Thank you for pointing this out and we apologize for the lack of clarity. We have rephrased this sentences as “Having found that LC activation resulted in a temporal rearrangement of the event timepoints in each VPm neuron’s spiking response to the same WGN tactile stimulus clip, we next investigated how ideal the temporal arraignment of the event structure each VPm neuron used to encode the WGN stimulus clip was with and without LC stimulation.

Our data also showed that spikes within those remaining events carried more information about stimulus features (Figure 3f), suggesting that the temporal code of a single remaining event was altered by LC stimulation.

Minor point 1 (Line 45): Please add a reference to this statement.

We have added a reference.

Minor Point 2 (Line 47): Please be more specific here, as there are distinct ion mechanisms leading to bust spiking in other neuron types.

Thank you for pointing this out. We have clarified the neuron type in that study.

Minor Point 3 (Line 55-60): Please consider to provide here clearer references to the relevant publication.

These are from our previous work. We have added the reference.

Minor point 4 (Line 99-101): Please be more specific here. Range of PB concentration, how do you monitor the anesthesia depth, approx. duration of the operation? Also add a statement about the means to minimize animal suffering.

We have added details about PB concentration and procedures to maintain a surgical level of anesthesia.

Minor Point 5 (Line 107): Providing stereotactic parameters may be helpful here. Do you perform histology to identify the correct location and the ration of LC neurons transfected?

We have provided stereotaxic parameters in the revised manuscript. We did not conduct histology in the new experiments due to COVID19 pandemic. But the histological confirmation of LC location and transfection has been included in the previous publication (Rodenkirch et al., Nat. Neurosci., 2019).

Minor Point 6 (Line 121): Please describe in more detail how you identify that the principal whisker was stimulated.

Thank you. We have added details about how to identify the principal whisker.

Minor point 7 (Line 131): Please add a reference here informing the reader that this information is a published result.

Done

Minor point 8 (Line 145): I miss a description of the exponent “T”.

We have defined the symbol “T”.

Minor Point 9 (Line 151): Please provide a definition for the tern “conditional feature”.

Sorry for the confusion. We have replaced “conditional feature” with “feature during LC activation”.

Minor Point 10 ( Line 160): I was misled by the term “directionality” at this point, assuming here that you also investigated the tuning map of the VPn neuron. Consider stating here that directionality just refers to forward-backward.

 We have clarified the directionality only refers to direction within the 2D deflection plane.

Minor Point 11 (Line 173): From Fig. 1 it seems that this deconvultion is symmetrical in time and may thus interfere with the precise detection of “event” onsets (as the deconvoluted SDF will depend on the maximal frequency). Please estimate whether this will substantially influence the feature detection.

Thank you for pointing this lack of clarity out. We have used the previously defined method of event detection using an adaptive boxcar kernel to smooth the SDF as published by Mainen, Z.F. & Sejnowski, T.J. Reliability of spike timing in neocortical neurons. Science. To test whether our event onset and offset points were fully capturing the entire spiking event we used different firing rate thresholds which effectively widen or shorten the temporal width of the events and we found the results were similar (supplementary figure 2).  We also tested the results with 6 or 10 ms padded to either side of the event and the results were qualitatively similar (not included in publication). Importantly, whenever performing information analysis we measure the information transmitted by the exact spike times that fell within each event, not the information carried by the onset of the event itself.  

Minor point 12 (Line 290): Sentence incomplete??

We apologize for the confusion. We have rephrased the sentence.

Minor Point 13 (Line 307): “we found that the feature selectivity [….] of spikes within emerged events was much improved as compared to spikes within removed events”. Please consider to use a comparative term like “higher”, as there can be no functional progression in these mutually exclusive events.

Corrected.

Minor Point 14 (Line 310): Shouldn't this be “about encoded features”, as the comparison between these events is not restricted to a predefined feature?

Thank you. The typo has been corrected.

Reviewer 3 Report

Comments and Suggestions for Authors

The manuscript biology-2809815 entitled "Optimization of Temporal Coding of Tactile Information in the Thalamus by Locus Coeruleus Activation", submitted by Charles Rodenkirch and Qi Wang, represents a very remarkable contribution to the knowledge of the processing of tactile information by the locus coeruleus at the thalamic level. The work is in line with previous studies by the same authors (doi: 10.1016/j.celrep.2017.08.094; doi:10.1038/S41593-018-580 0283-1).

The manuscript is well-written, easy to follow in its development, and well-referenced. The material and methods are well detailed in such a way as to make the experiments reproducible. The results and discussion are correct.

I would just like to suggest to the authors that they should include the animal model in the title so as not to create confusion or think that it is a general revision. Also, to homogenize the edition, in lines 377-378 replace full authors reference with Petersen et al. or Petersen and co-workers.

Author Response

Reviewer 3

The manuscript biology-2809815 entitled "Optimization of Temporal Coding of Tactile Information in the Thalamus by Locus Coeruleus Activation", submitted by Charles Rodenkirch and Qi Wang, represents a very remarkable contribution to the knowledge of the processing of tactile information by the locus coeruleus at the thalamic level. The work is in line with previous studies by the same authors (doi: 10.1016/j.celrep.2017.08.094; doi:10.1038/S41593-018-580 0283-1).

The manuscript is well-written, easy to follow in its development, and well-referenced. The material and methods are well detailed in such a way as to make the experiments reproducible. The results and discussion are correct.

I would just like to suggest to the authors that they should include the animal model in the title so as not to create confusion or think that it is a general revision. Also, to homogenize the edition, in lines 377-378 replace full authors reference with Petersen et al. or Petersen and co-workers.

Thank you for your support. We have revised the manuscript as suggested.